# LLMs Fail to Recognize Mathematical Insolvability

## Abstract

While modern large language models (LLMs) achieve high accuracy on many challenging math benchmarks, they often struggle to recognize the insolvabilty of ill-posed problems. However, existing benchmarks for insolvable problems are either modified from elementary-level math questions or lack rigorous validation of their insolvability. There is a lack of benchmarks featuring inherently insolvable problems that require deep mathematical knowledge to identify their insolvability. To fill the gap, we introduce *MathTrap300*, the first benchmark consisting of 300 insolvable, ill-posed math problems with fundamental mathematical contradictions that demand deep domain knowledge to detect. In this work, we manually derived these problems from their well-posed counterparts through careful modifications and rigorous verification of ill-posedness by PhD-level experts. We then present a fine-grained, three-stage LLM judge framework, designed from the observation of LLMs' responses to insolvable problems. The framework captures signals from both final answers and intermediate reasoning, providing richer metrics and enabling a more faithful assessment of insolvability recognition. Our evaluation of recent advanced LLMs on MathTrap300, combined with a detailed analysis of their response patterns, reveals a clear drop in accuracy from well-posed problems to their insolvable counterparts. Common failure modes are categorized into hallucination, guessing, and neglect of conditions, etc. It is also observed that even when models recognize the insolvabilty, they still attempt to force a solution, a form of sycophantic behavior. Dataset and the evaluation code are available

## 1 Introduction

Large language models (LLMs) have made rapid and substantive progress on mathematical reasoning, achieving high accuracies on widely used benchmarks such as MATH (Hendrycks et al., 2021), AMC or AIME, especially after the emergence of reinforcement learning to incentivize LLM reasoning (Guo et al., 2025; Jaech et al., 2024). However, despite the impressive performance on well-posed math datasets, modern LLMs still struggle with identifying the insolvability when they face ill-posed math questions. In practice, a confident but spurious "solution" can be more harmful than abstention, so the capabiltiy of identifying insolvability is essential. Most current math benchmarks primarily target on using increasingly hard, well-posed problems to guide the improvement of LLM reasoning, but less benchmarks focus on measuring LLMs' performance on insolvable problems.

There are several previous works on insolvable math benchmarks (Zhao et al., 2024; Ma et al., 2025; Tian et al., 2024; Xue et al., 2025). Nevertheless, these efforts exhibit several common limitations. First, many of the problems are derived from elementary-level mathematics (Zhao et al., 2024; Tian et al., 2024; Ma et al., 2025), resulting in low difficulty that fails to challenge contemporary LLMs, especially reasoning models. Second, some datasets (Xue et al., 2025) are adapted from competition-level problems, but the automatic modification pipeline via LLMs makes a substantial portion of problems actually still solvable,

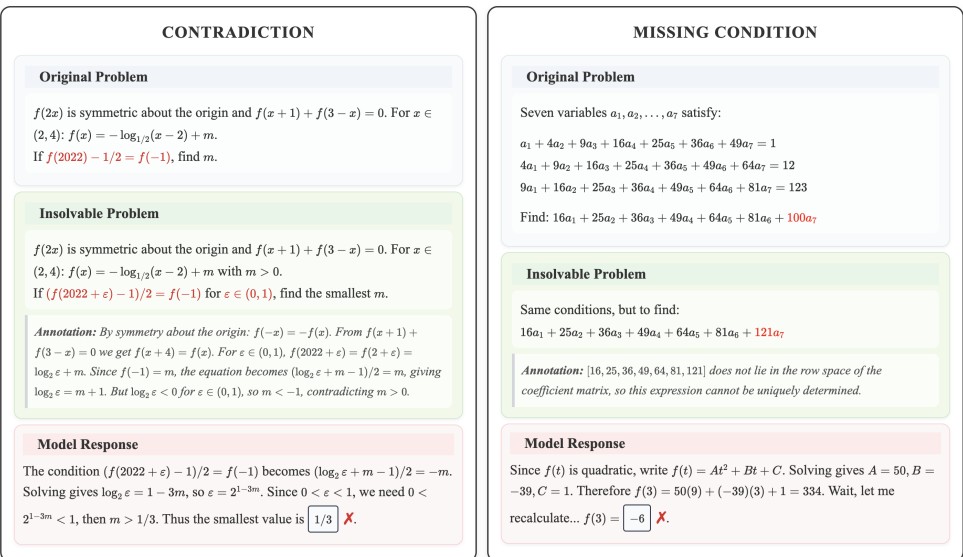

Figure 1: A demonstration of insolvable problems derived from original problems with rigorous mathematical insolvability, and corresponding model responses. Despite the insolvability, both responses cannot recognize it and force final answers. **Left:** Contradiction - the insolvable problem creates mathematical inconsistency adding another variable $\epsilon$ whose range precludes any positive $m$. **Right:** Missing Condition - the insolvable problem lacks sufficient constraints by changing the target coefficient from 100 to 121.

or non-mathematical traps. Finally, LLMs often show complex behaviors when face insolvable problems, while symbolic or string-matching based methods for assessment Xue et al. (2025) are too rigid to capture such nuanced responses, and LLM judges with single binary metrics cannot cannot disentangle the diverse failure modes of LLMs.

To address these gaps, we introduce **MathTrap300**, a manually curated, double-verified dataset of 300 inherently insolvable mathematical problems with missing or contradictory conditions. Two sample problems are shown in 1. The difference between our work and previous works are shown in Table 1.

| Method | Difficulty | Verified Insolvability | Fair Assessment | Pattern Analysis |
|---|---|---|---|---|
| MathTrap (Zhao et al., 2024) | ✗ | ✓ | ● | ✗ |
| PMC (Tian et al., 2024) | ✗ | ✗ | ● | ✗ |
| UMP (Ma et al., 2025) | ✗ | ✗ | ● | ● |
| ReliableMath (Xue et al., 2025) | ✓ | ✗ | ✗ | ● |
| **MathTrap300 (Ours)** | ✓ | ✓ | ✓ | ✓ |

Table 1: **Compact comparison with previous works.** Legend: ✓ good, ● limited, ✗ poor.

Specifically, our contributions are:

1. **Inherent mathematical insolvability with quality controls:** MathTrap300 contains 300 problems crafted and double-verified by PhD-level experts. We implement paraphrasing to mitigate training contamination and memorization effect (Huang et al., 2025).

2. **Three-stage judge pipeline:** Beyond binary scoring, we decompose model behavior into three components: final-answer claim of insolvability, midway identification, and problem modification. Each component is assessed by LLM judges with sophisticated procedure to construct few-shot prompts. The reliability of the judge system is verified by the alignment of human judge.

3. **Benchmarking with detailed pattern analysis:** We benchmark 28 state-of-the-art models on MathTrap300 and report the diverse metrics using our LLM judge. A detailed pattern analysis is conducted to classify the common behaviors of LLM response when facing insolvable problems. A correlation of sycophancy behaviors and capability of handling insolvability is proposed based on our observation.

MathTrap300 establishes a rigorous benchmark for evaluating LLMs' ability to recognize mathematical impossibility, providing essential insights for developing more reliable mathematical reasoning systems.

## 2 RELATED WORKS

### 2.1 MATH REASONING BENCHMARKS

To evaluate LLMs' reasoning capability, numerous math reasoning benchmarks have been proposed. Representative ones include GSM8K (Cobbe et al., 2021), MATH (Hendrycks et al., 2021), Minerva (Lewkowycz et al., 2022), OlympiadBench (He et al., 2024), AMC, and AIME. As LLM reasoning keeps improving, the community expect higher difficulty of math benchmarks to avoid saturation. For some recently released models such as Grok-4 and GPT-5, their accuracies on AIME has already exceeded 90%.

### 2.2 VARIANTS OF REASONING BENCHMARKS

In addition to proposing harder math benchmarks, another line is creating the variants based on existing benchmarks to assess data contamination, memorization, or robustness against irrelevant information. GSM-IC (Shi et al., 2023) introduces irrelevant descriptions into GSM8K to test a model's sensitivity to distracting information, which shows significant performance drop. Functional MATH (Srivastava et al., 2024) modifies the problems of MATH into a dynamic, functional benchmarks. MATH$^2$ (Shah et al., 2024) extracts the requried skills of two random problems from MATH, and combine them to generate a harder problem requiring compositional skills. Similarly, GSM-Symbolic (Mirzadeh et al., 2024) created a symbolic template of GSM8K, where the numeric values, names etc. can changed by programming, and great data contamination is observed across a series of models. MATH-Perturb (Huang et al., 2025) introduces simple and hard perturbation into MATH by minimal editing, where the solution logic keeps unchanged for the former and fundamentally changed for the later. Memorization is observed where models try to follow the paths of original problems.

### 2.3 BENCHMARKS FOR INSOLVABLE MATH PROBLEMS

MathTrap (Zhao et al., 2024) one of the pioneering work on insolvable problems. However, the dataset only has around 150 problems and mostly derived from elementary-level problems. The single LLM-based judge only limits diagnostic resolution. Another concurrent pioneering work is UMP, (Ma et al., 2025) where, similarly, most questiosn are elementary-level and the "unreasonability" automatically introduced by LLMs are more about commonsense-level contradictions rather than math. PMC (Tian et al., 2024) automatically converts arithmetic problems at scale (around 5000) into missing-condition and contradictory-condition variants, but still most are easy questions and automatic generation makes the insolvability superficial. ReliableMath (Xue et al., 2025) is the first one to use competition-level math problems in insolvable math benchmarks, but the automatic pipeline yields "pseudo-insolvability", where the removed / inserted

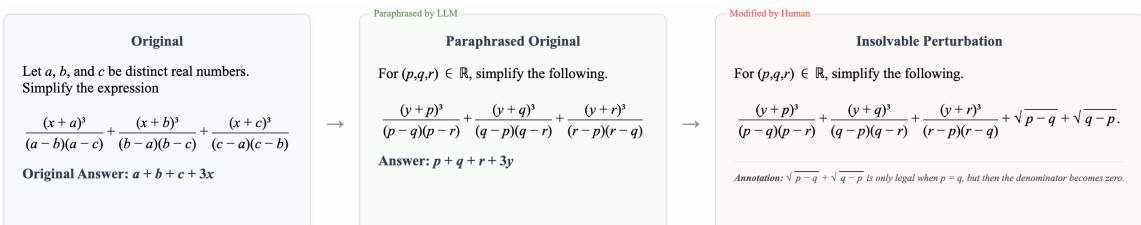

Figure 2: **Insolvable problem construction process.**

information does not impact the solvability. In addition, ReliableMath uses the appearance of "unsolvable" in final answer as the criterion, leading to underestimated accuracies.

Examples of these problematic questions from existing benchmarks can be found in Appendix A.

# 3 METHODS

## 3.1 DATASET CONSTRUCTION

MathTrap300 collects problems from MATH, AIME 2025, AMC, Chinese high school and selective admission exams, and original items authored by the authors. In addition, we adapted 37 questions from MathTrap (Zhao et al., 2024) to ensure sufficient discriminative power. We restrict attention to *well-stated* problems and exclude ambiguity classes that arise from vague wording. We modify the problems so that the resultant questions are either contradictory, where the conditions in the problem cannot simultaneously exist, or missing conditions, where the given information is insufficient to determine the desired values. Figure 1 gives concrete examples of these two insolvability types. In total, MathTrap300 contains 274 Contradiction problems and 26 Missing Condition problems.

Following (Huang et al., 2025), we observed strong memorization effects of perturbed insolvable questions and therefore paraphrase original problems with an LLM before inserting subtle insolvability edits: paraphrases follow strict renaming and notation-preservation rules so they preserve content and difficulty while reducing lexical overlap. For quality control we recruited domain-qualified reviewers (strong recent high-school competitors, STEM undergraduates, and PhD-level researchers); problems were required to confuse competitive screening models (e.g., o4-mini) during triage—these screening models are explicitly excluded from final benchmarks to avoid bias—and each item receives an LLM proofreading pass (clarity and insolvability checks) followed by authors' review.

## 3.2 EVALUATION METHODS

For the accuracy of original problems, we directly compare the answer generated by LLMs with the ground-truth answer. First the symbolic comparison library adopted from Shao et al. (2024) is adopted. In ordre to handle the answers that cannot be recognized by symbolic comparison, answers that is concluded as incorrect by symbolic comparison will be feed into a LLM to compare the equivalence. Kimi K2 is used and the prompt can be seen in supplementary materials.

## 3.3 THREE-STEP LLM JUDGE

For problems intentionally designed to be insolvable, we evaluate models with a three-step judge pipeline. Prior work has typically relied on a single judge or binary criterion, which can overlook the diverse ways

in which models signal awareness of contradictions. Our approach, inspired by earlier work on multi-stage evaluation of mathematical reasoning (Sheng et al., 2025), extends this idea to specifically capture the nuanced ways models might recognize insolvability. Unlike binary classification, this framework credits partial recognition in intermediate reasoning steps, thereby providing a more comprehensive assessment of models' mathematical awareness, as illustrated in Figure 3.

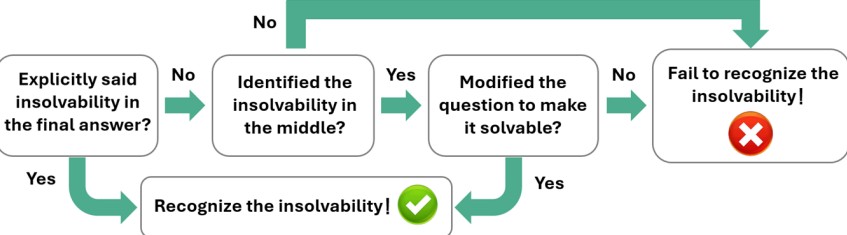

Figure 3: **Decision pipeline for insolvability recognition.** A model response is passed through three sequential judges (final-answer, identification, modification). The insolvability is recognized if either the final answer declares insolvability, or if the model both identifies the issue in its reasoning and proposes a reasonable modification.

1. **Final-answer judge.** Does the final answer assert insolvability (e.g., "no solution", "insufficient information") ? A correct final answer for an insolvable problem is to explicitly declare it insolvable. Otherwise, the response is passed to the next judge

2. **Midway-identification judge.** Is the model aware the nature of the insolvability in its intermediate reasoning (e.g., "constraints $A$ and $B$ contradict because . . . ", or "there seems to be a typo here").

3. **Problem-modification judge.** If the model recognizes the issue, does it attempt a reasonable modification to the problem that would make it solvable (for example, adding a plausible missing constraint, or fixing a variable domain).

**Decision rule.** Let $J_1$, $J_2$, and $J_3$ denote the binary outputs of the final-answer, identification, and problem-modification judges, respectively. A model is considered to recognize the insolvability if:

$$\text{Recognition} = J_1 \vee (J_2 \wedge J_3)$$

To illustrate how these rules manifest in practice, Figure 4 provides concrete examples of acceptable and unacceptable behaviours for both Contradiction and Missing Condition insolvabilities.

| Model | Accuracy | Precision | Recall | F1-score |
|---|---|---|---|---|
| Final Answer | 1.00 | 1.00 | 1.00 | 1.00 |
| Midway Identification | 0.92 | 0.88 | 1.00 | 0.94 |
| Problem Modification | 0.74 | 0.61 | 1.00 | 0.76 |
| Overall Assessment | 0.79 | 0.72 | 1.00 | 0.84 |

Table 2: Performance metrics derived from confusion matrices.

To assess the reliability of our judge pipeline, we constructed confusion matrices for each of the three judges (final-answer, midway-identification, and problem-modification) and compared them against human labels; implementation and tuning details appear in the Supplementary Materials.

**CONTRADICTION**

**Insolvable Problem**

$f(2x)$ is symmetric about the origin and $f(x+1) + f(3-x) = 0$. For $x \in (2,4)$: $f(x) = -\log_{1/2}(x-2) + m$. If $(f(2022+\varepsilon) - 1)/2 = f(-1)$ for $\varepsilon \in (0,1)$, find the smallest value of $m$.

✓ **Correct Answer 1:**

There is no positive m that satisfies the conditions.

**[explicitly states the insolvability in the final answer]**

✓ **Correct Answer 2:**

There seems to be an inconsistency, since no choice of $\varepsilon \in (0,1)$ exists for positive $m$ **[realizing the insolvability midway]**. If the intended range of $\varepsilon$ is $\varepsilon > 1$, then we get the minimum $m = 1$ **[modifying the problem]**

**MISSING CONDITION**

**Insolvable Problem**

Consider seven variables $a_1, a_2, ..., a_7$ satisfying:

$a_1 + 4a_2 + 9a_3 + 16a_4 + 25a_5 + 36a_6 + 49a_7 = 1$

$4a_1 + 9a_2 + 16a_3 + 25a_4 + 36a_5 + 49a_6 + 64a_7 = 12$

$9a_1 + 16a_2 + 25a_3 + 36a_4 + 49a_5 + 64a_6 + 81a_7 = 123$

**Find:** $16a_1 + 25a_2 + 36a_3 + 49a_4 + 64a_5 + 81a_6 + 121a_7$

✓ **Correct Answer 1:**

The desired expression cannot be uniquely determined.

**[explicitly states the insolvability in the final answer]**

✓ **Correct Answer 2:**

$16a_1 + 25a_2 + ... + 121a_7 = 334 + a_7$ leaves one free variable, so the value cannot be uniquely determined **[realizing the insolvability midway]**. If we set $a_7 = 0$, then the problem becomes solvable, with the answer $334$ **[modifying the problem]**.

Figure 4: **Acceptable model behaviours.** For Contradiction problem, a correct response either explicitly states the inconsistency or proposes a natural correction. For Missing Condition problem, a correct response either states the underdetermination or adds a natural constraint to make the problem well-posed.

Among the three judges, the final-answer judge exhibits the strongest alignment with human annotations, achieving perfect precision and recall (1.00). The midway-identification judge improves coverage by capturing intermediate contradiction signals, though at the cost of a modest drop in precision (0.88). By contrast, the problem-modification judge attains high recall (1.00) but substantially lower precision (0.61), indicating that many model-proposed fixes judged positive by the pipeline are not judged reasonable by human annotators. Overall, the combined decision rule yields an $F_1$ score of about **84%**. Unlike prior work that typically relies on a single, rigid judging criterion, which leads to systematic underestimation of overall recognition accuracy, our multi-judge scheme provides a more balanced and arguably fairer estimate. In fact, since our rule admits high recall (and thus allows in more false positives), it may even overestimate performance, but we view this as reasonable: the rule reflects models' ability to detect contradictions at any reasoning step and to attempt plausible fixes, which is a more faithful measure of their practical sensitivity to insolvability.

## 4 EXPERIMENTAL RESULTS

### 4.1 LLMs PERFORMANCE

To ensure comparability, we therefore restrict evaluation to non-empty responses only. With the judge pipeline validated, we evaluate a broad set of LLMs on both the original problems and our insolvable edits. Table 4.1 reports benchmark accuracy.Because some models (e.g., Grok, OpenAI) have very long token outputs, we set the maximum generation length to 21,000 tokens. However, when such models exhaust this budget prematurely during the thinning stage, they may terminate with an empty response.

Table 3: Benchmark accuracy (non-empty responses)

| Model | Final Ans. Acc | Identify Acc | Overall Acc | Original Acc | Acc.Drop |
|---|---|---|---|---|---|
| *Open Chat Models* | | | | | |
| DeepSeek-V3.1 | 9.33 | 84.00 | 80.67 | 84.67 | 4.00 |
| Kimi-K2-Instruct-0905 | 19.33 | 70.33 | 65.00 | 82.00 | 17.00 |
| Llama3.3-70B-Instruct | 1.33 | 33.00 | 28.67 | 67.33 | 38.67 |
| Llama4-Maverick-Instruct | 4.67 | 53.33 | 46.67 | 77.00 | 30.33 |
| Llama4-Scout-Instruct | 1.67 | 44.00 | 37.33 | 76.67 | 39.33 |
| Qwen2.5-72B-Instruct | 3.00 | 38.67 | 34.33 | 67.67 | 33.33 |
| Qwen3-235B-A22B-Instruct-2507 | 8.67 | 93.67 | 90.33 | 88.00 | -2.33 |
| Qwen3-30B-A3B-Instruct-2507 | 7.33 | 90.33 | 87.33 | 86.33 | -1.00 |
| *Open-source Reasoning LLMs* | | | | | |
| DeepSeek-reasoner | 9.33 | 96.33 | 88.00 | 94.67 | 6.67 |
| gpt-oss-120b | 26.67 | 92.33 | 90.33 | 87.67 | -2.67 |
| gpt-oss-20b | 30.00 | 93.67 | 89.00 | 87.33 | -1.67 |
| Phi-4-reasoning-plus | 4.00 | 32.33 | 28.67 | 64.33 | 35.67 |
| Qwen3-235B-A22B-Thinking-2507 | 6.33 | 93.33 | 79.00 | 91.00 | 12.00 |
| Qwen3-30B-A3B-Thinking-2507 | 8.67 | 94.67 | 90.67 | 92.00 | 1.33 |
| Qwen3-8B (Thinking) | 5.33 | 95.33 | 87.67 | 80.67 | -7.00 |
| Qwen2.5-Math-72B-Instruct | 3.67 | 39.33 | 35.67 | 70.00 | 34.33 |
| *Proprietary Chat LLMs* | | | | | |
| Gemini 2.5 Flash (no thinking) | 23.00 | 89.67 | 83.67 | 89.33 | 5.67 |
| Claude Sonnet 4 (no thinking) | 6.00 | 69.00 | 64.00 | 78.67 | 14.67 |
| gpt-4o-2024-11-20 | 18.00 | 28.33 | 27.00 | 61.67 | 34.67 |
| gpt-4.1-2025-04-14 | 24.33 | 75.67 | 72.67 | 76.67 | 4.00 |
| *Proprietary Reasoning LLMs* | | | | | |
| claude-sonnet-4 (extended thinking) | 11.67 | 90.00 | 83.67 | 86.33 | 2.67 |
| Gemini 2.5 Flash (with thinking) | 27.00 | 93.67 | 88.33 | 87.33 | -1.00 |
| Gemini 2.5 Pro | 20.00 | 91.67 | 85.33 | 91.33 | 6.00 |
| Grok-4 | 19.05 | 70.13 | 68.40 | 95.89 | 27.49 |
| Grok-3-mini-beta | 6.00 | 91.67 | 81.67 | 88.67 | 7.00 |
| o3-2025-04-16 | 33.11 | 77.26 | 74.58 | 92.33 | 17.75 |
| o4-mini-2025-04-16 | 30.00 | 64.33 | 63.00 | 90.00 | 27.00 |
| gpt-5-2025-08-07 | 45.15 | 85.28 | 83.61 | 90.67 | 7.05 |

**Trap vs. Original Accuracy.** Accuracy on trap instances is substantially lower than on original problems for almost all models. Even the strongest systems (e.g., GPT-5, o3, Gemini 2.5 Pro) show nontrivial drops (4–18%), while weaker open-chat baselines such as Llama-3.3, Llama-4, or Qwen-2.5 Instruct models suffer losses exceeding 30%. This confirms that our trap edits introduce genuine difficulty rather than superficial perturbations. Interestingly, a handful of large reasoning models (e.g., GPT-OSS-120B, Qwen-3 Instruct/Thinking) report negative drops, meaning their trap accuracy actually surpasses performance on original items—likely reflecting robustness to noisy or underspecified conditions.

**Reasoning vs. Chat Models.** Reasoning-oriented models clearly outperform general-purpose chat models, especially on the identify and overall recognition metrics. Open reasoning systems like Qwen-3-30B-Thinking (90.7% overall) and GPT-OSS-20B/120B ($\approx$ 90% overall) consistently surpass their chat counterparts with the same or larger parameter counts. Proprietary reasoning systems (GPT-5, o3, Claude-Sonnet-

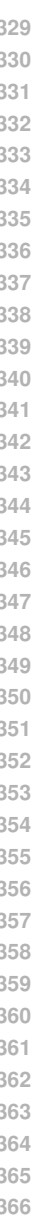

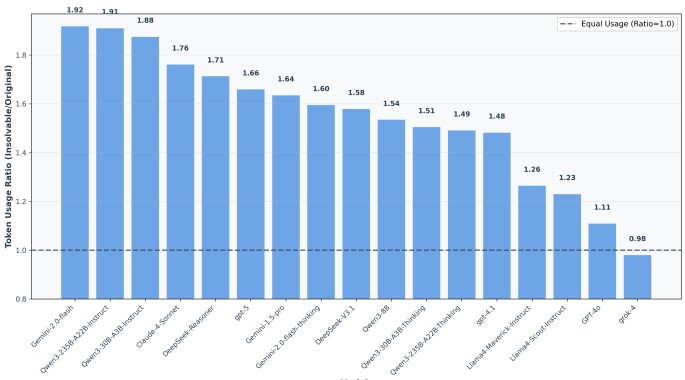

Figure 5: Token Usage Ratio Comparison (Insolvable vs Original Problems).

extended, Gemini-Flash-thinking) also achieve strong recognition, often exceeding 80 overall, while their paired chat models fall 10–20 % lower. This gap highlights the importance of deliberate step-by-step reasoning traces in surfacing contradictions, whereas chat-style models often generate fluent but overconfident responses that ignore hidden inconsistencies.

## 5 FAILURE MODES ANALYSIS

**Guess** The model's reasoning traces often contain tentative checks or partial diagnostics, suggesting some awareness of inconsistency. However, it rarely proceeds to an explicit declaration of insolvability or to a minimal corrective reformulation. Instead, it cycles through heuristic approaches or concludes with a numerical answer lacking derivational support (see Case 1 in Figure 6). Such outputs are typically accompanied by hedging expressions and brief diagnostic cues, but without a substantiated rationale. Likely contributing factors include a systemic preference for producing determinate answers over abstention, insufficient prompting for impossibility certification, and truncated reasoning that prevents full verification. In evaluation, these cases artificially inflate apparent competence on final-answer metrics, as superficially confident numbers mask unresolved uncertainty in the underlying reasoning, which corresponds to the judge output $(J_1, J_2, J_3) = (0, 1, 0)$.

**Hallucinate** The model produces intermediate steps, lemmas, or algebraic manipulations that appear valid but lack justification. Typical instances include fabricated identities, unjustified substitutions, or rule applications outside their valid scope, which yield fluent yet invalid derivations (see Case 2). Such outputs are often expressed with confidence and fluency, making errors difficult to detect without careful checking. Contributing factors likely include training objectives that emphasize persuasive natural language over formal correctness and the absence of built-in symbolic reasoning or step-level verification. In evaluation, these behaviors can mislead superficial checks and inflate estimates of model competence.

**Constraint-ignore** The model fails to enforce domain constraints, such as positivity, integrality, distinctness, or interval bounds—and continues with derivations that render the solution invalid for the original problem (see Case 3). Typical signs include final answers lying outside the stated domain or reasoning traces that never enumerate or check the relevant conditions, which typically produce $(0, 0, 0)$. Likely contributing factors are over-reliance on algebraic templates without constraint-checking mechanisms and limited training exposure to constraint-sensitive examples. In evaluation, these outputs may be scored as correct despite being logically invalid, leading to overestimation of model performance.

Across the three failure modes, models also exhibit two common fallback behaviors when faced with difficult instances. First, they resort to exhaustive case enumeration or brute-force testing rather than producing a concise analytic diagnosis. Second, they expand reasoning into lengthy, low-level arithmetic computations that increase verbosity and raise the likelihood of basic computational errors. These behaviors lead to long, repetitive reasoning traces that obscure the central logical claims, inflate token usage, and amplify arithmetic mistakes. As a result, automated judges are less effective, and genuine diagnostic signals become harder to detect. Mitigation strategies include discouraging unnecessary enumeration or redundancy (e.g., through brevity or efficiency penalties), requiring concise justification before permitting extensive casework, and automatically compressing or extracting key claims from verbose reasoning so that downstream symbolic checks can focus on the essential steps.

In summary, the judge-validated taxonomy indicates that larger reasoning-oriented models with longer chain-of-thought traces are more effective at detecting and diagnosing unsolvable problems, whereas many models still default to guessing, fabricating steps, or brute-force enumeration.

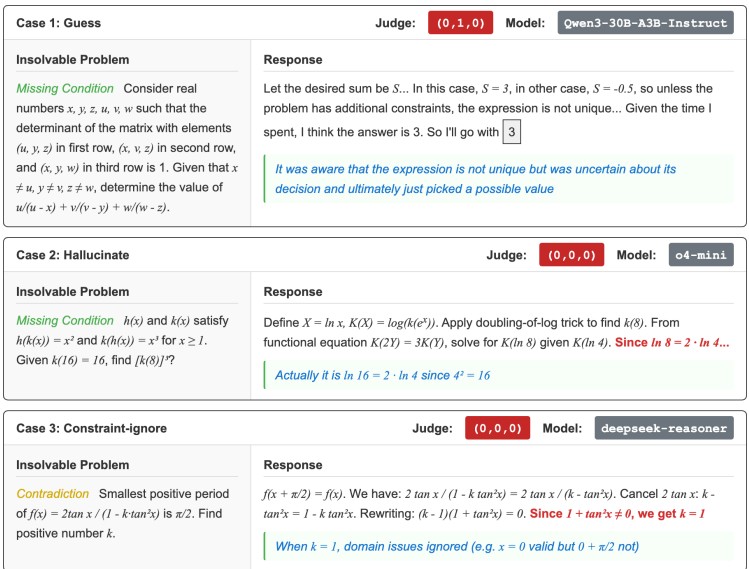

Figure 6: **Failure Pattern Analysis**

## 6 CONCLUSION

We conclude that MathTrap300, a curated set of 300 expert-verified insolvable math problems, together with our three-stage LLM judge reveals a clear reliability gap: modern LLMs show a marked drop in performance on trap instances and do not reliably detect deep mathematical insolvability. Crucially, failure to recognize insolvability is strongly correlated with sycophantic forcing of answers - models that produce fluent, accommodating outputs are far less likely to report "no solution" and frequently emit pseudo-solutions even when their own intermediate reasoning signals a contradiction. We release MathTrap300 and the judge code to support follow-up research.

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

# A   DETAILED ANALYSIS OF PRIOR WORK LIMITATIONS

---

**Triangular Numbers Problem**                                    `ReliableMath`

|  |  |
|---|---|
| **Original Problem** | **Modified Question** |

A triangular number is a positive integer that can be expressed in the form $t_n = 1+2+3+\cdots +n$, for some positive integer $n$. The three smallest triangular numbers that are also perfect squares are $t_1 = 1 = 1^2$, $t_8 = 36 = 6^2$, $t_{49} = 1225 = 35^2$. What is the sum of the digits of the fourth smallest triangular number that is also a perfect square?

A triangular number is a positive integer (the definition is removed.) The three smallest triangular numbers that are also perfect squares are $t_1 = 1 = 1^2$, $t_8 = 36 = 6^2$, $t_{49} = 1225 = 35^2$. What is the sum of the digits of the fourth smallest triangular number that is also a perfect square?

**Design Limitations**

*Removing the triangular number definition may not create genuine insolvability since LLMs have extensive training on mathematical concepts. The background knowledge compensates for the missing explicit definition.*

---

**Salary Distribution Problem**                                    `UMP`

|  |  |
|---|---|
| **Original Problem** | **Modified Question** |

Zaid's $6000 salary: 2/3 rent, **1/4** of rest donated, $700 to daughter. What's left?

Zaid's $6000 salary: 2/3 rent, **3/4** of rest donated, $700 to daughter. What's left?

**Design Limitations**

*The direct change from 1/4 to 3/4 creates an obvious mathematical constraint violation that may be too straightforward to effectively challenge or mislead advanced language models.*

---

Figure 7

**Jelly Bean Counting Problem** PMC

| Original Problem | Modified Question |
|---|---|

**Original Problem**

Gunter is trying to count the jelly beans in a jar. He asks his friends how many they think are in the jar. One says 80 . Another says 20 more than half the first one. A third says 25% more than the first one. What is their average guess?

**Modified Question**

**Type 1 (Missing Condition):**
Gunter is trying to count the jelly beans in a jar. He asks his friends how many they think are in the jar. One says a certain number . Another says 20 more than half the first one. A third says 25% more than the first one. What is their average guess?

**Type 2 (Contradiction):**
Gunter is trying to count the jelly beans in a jar. He asks his friends how many they think are in the jar. One says 80 . Another says 20 more than half the first one. A third says 25% more than the first one. What is their average guess if the second friend's guess is 50 ?

**Design Limitations**

*Both humans and LLMs naturally follow conditional "if" statements and treat "certain number" as variable x. PMC's template-based approach for introducing missing conditions and contradictions is straightforward but may be too formulaic to challenge advanced models.*

Figure 8

# B EXPERIMENTAL DETAILS

## B.1 EXPERIMENTAL SETUPS

Table 4: List of LLMs with their default parameters

| Model | Size (B) | Temp. | Top_p | Top_k | Max_tokens | Reasoning effort |
|---|---|---|---|---|---|---|
| *Open Chat Models* | | | | | | |
| DeepSeek-V3.1 | 671 | 0 | N/A | N/A | 20000 | |
| Kimi-K2-Instruct-0905 | 1000 | 0.6 | 0.95 | N/A | 20000 | |
| Llama3.3-70B-Instruct | 70 | 0.6 | 0.95 | 20 | 20000 | |
| Llama4-Maverick-Instruct | 402 | 0 | N/A | N/A | 20000 | |
| Llama4-Scout-Instruct | 109 | 0 | N/A | N/A | 20000 | |
| Qwen2.5-72B-Instruct | 72 | 0.6 | 0.95 | 20 | 20000 | |
| Qwen3-235B-A22B-Instruct-2507 | 238 | 0.7 | 0.8 | 20 | 20000 | |
| Qwen3-30B-A3B-Instruct-2507 | 30 | 0.7 | 0.8 | 20 | 20000 | |
| *Open-source Reasoning LLMs* | | | | | | |
| DeepSeek-V3.1-Think | 671 | 0 | N/A | N/A | 20000 | |
| GPT-oss-120b | 120 | 1 | 1 | N/A | 20000 | |
| GPT-oss-20b | 20 | 1 | 1 | N/A | 20000 | |
| Phi-4-reasoning-plus | 14.7 | 0.8 | 0.95 | 50 | 20000 | |
| Qwen3-235B-A22B-Thinking-2507 | 238 | 0.6 | 0.95 | 20 | 20000 | 18000 |
| Qwen3-30B-A3B-Thinking-2507 | 30 | 0.6 | 0.95 | 20 | 20000 | 18000 |
| Qwen3-8B (Thinking) | 8 | 0.6 | 0.95 | 20 | 20000 | 18000 |
| Qwen2.5-Math-72B-Instruct | 72 | 0 | N/A | N/A | 4096 | |
| *Proprietary Chat LLMs* | | | | | | |
| Gemini 2.5 Flash (no thinking) | – | 1 | 0.95 | 64 | 20000 | |
| claude-sonnet-4-20250514 (no thinking) | – | 1 | 0.95 | N/A | 20000 | |
| GPT-4o-2024-11-20 | – | 0.6 | 0.95 | N/A | 16384 | |
| GPT-4.1-2025-04-14 | – | 0.6 | 0.95 | N/A | 20000 | |
| *Proprietary Reasoning LLMs* | | | | | | |
| claude-sonnet-4-20250514 (extended thinking) | – | 1 | 0.95 | N/A | 20000 | 18000 |
| Gemini 2.5 Flash (with thinking) | – | 1 | 0.95 | 64 | 20000 | 18000 |
| Gemini 2.5 Pro | – | 1 | 0.95 | 64 | 20000 | 18000 |
| grok-4 | – | 0.6 | 0.95 | N/A | 20000 | |
| grok-3-mini-beta | – | 0.6 | 0.95 | N/A | 20000 | high |
| o3-2025-04-16 | – | N/A | N/A | N/A | 20000 | medium |
| o4-mini-2025-04-16 | – | N/A | N/A | N/A | 20000 | medium |
| GPT-5-2025-08-07 | – | N/A | N/A | N/A | 20000 | medium |

## B.2 FAILURE PATTERN ANALYSIS TRAP PROBLEM ANNOTATION

---

**Insolvable Problem 1: Matrix Determinant**

| Insolvable Problem | Annotation |
|---|---|
| *Missing Condition* Consider real numbers $x, y, z, u, v, w$ such that the determinant of the matrix with elements $(u, y, z)$ in first row, $(x, v, z)$ in second row, and $(x, y, w)$ in third row is 1. Given that $x \neq u, y \neq v, z \neq w$, determine the value of $u/(u - x) + v/(v - y) + w/(w - z)$. | We are given that the determinant equals $uvw - uyz - vxz - wxy + 2xyz = 1$. We fix $x = 0, y = 0, z = 1$. Then the determinant reduces to $uvw = 1$. The desired sum becomes $u/(u - 0) + v/(v - 0) + w/(w - 1) = u + v + w/(w - 1) = 3 + 1/(w - 1)$. You can choose any $w$ other than 0 and 1 to change the value of the desired formula, and you always can find valid $u, v$ that can satisfy the given constraints. |

---

**Insolvable Problem 2: Functional Equations**

| Insolvable Problem | Annotation |
|---|---|
| *Missing Condition* $h(x)$ and $k(x)$ satisfy $h(k(x)) = x^2$ and $k(h(x)) = x^3$ for all $x \geq 1$. Given that $k(16) = 16$, what is the value of $[k(8)]^3$? | Applying $k()$ to the first condition, and substituting $x = k(x)$ into the second condition, we can obtain $k(x^2) = [k(x)]^3$. From $k(16) = 16 = [k(4)]^3$ we get $k(4) = \sqrt[3]{16}$. Similarly, we get $k(2) = 16^{(1/9)}$. However, $8$ is not a perfect square, so the value $k(8)$ (and hence $[k(8)]^3 = k(64)$) is not determined. |

---

**Insolvable Problem 3: Periodic Function**

| Insolvable Problem | Annotation |
|---|---|
| *Contradiction* The smallest positive period of the function $f(x) = 2\tan x/(1 - k \cdot \tan^2 x)$ is $\pi/2$, find the value of the positive number $k$. | The domain of $f$ excludes $x = \pi/2 + n\pi, n \in \mathbb{Z}$, where $\tan x$ blows up. Since $f$ is invariant under a shift of $\pi/2$, one must also make $x = n\pi, n \in \mathbb{Z}$ out of domain. However, no matter how we choose $k$, we cannot make $x = n\pi, n \in \mathbb{Z}$ out of domain. |

Figure 9: **Failure Pattern Analysis Trap Problem Annotation**