# OpenReview forum: "LLMs Fail to Recognize Mathematical Unsolvability"
_ICLR.cc/2026/Conference — ICLR 2026 Conference Withdrawn Submission_

### Official Review · Reviewer_8G65 · 2025-10-27

**Soundness:** 2
**Presentation:** 3
**Contribution:** 2
**Rating:** 4
**Confidence:** 3

**Summary:**

This paper introduces MathTrap300, a curated benchmark of 300 mathematical problems that are inherently insolvable—either through contradictions or missing conditions. Unlike earlier datasets that focus on solvable or artificially modified elementary problems, MathTrap300 is designed to challenge LLMs with deeply ill-posed problems derived from verified competition-level sources.
The authors also propose a three-stage LLM-judge framework to assess models’ awareness of insolvability through: 1, Final-answer recognition; 2, Midway identification; 3, Problem modification.

Extensive experiments on 28 state-of-the-art models reveal significant performance drops when models confront ill-posed tasks. The study identifies three dominant failure modes—guessing, hallucination, and constraint ignorance—and correlates them with sycophantic answer-forcing tendencies.

**Strengths:**

The paper introduces MathTrap300, a first-of-its-kind dataset explicitly designed around inherent mathematical insolvability — not merely “tricky” or perturbed solvable problems. Prior datasets (e.g., MathTrap, UMP, PMC, ReliableMath) suffer from superficial or automatically generated “unsolvable” problems that often remain solvable upon inspection. This work uniquely focuses on logical contradictions and missing-condition problems that are genuinely unsolvable under formal reasoning.

**Weaknesses:**

- With only 300 problems, MathTrap300 may be too small to serve as a statistically robust or comprehensive benchmark, especially considering the vast diversity of mathematical domains and types of insolvability.

- The paper relies entirely on manual, expert-driven construction, which is high-quality but not scalable. It does not discuss how the benchmark could be systematically extended—e.g., via semi-automated pipelines using symbolic solvers, theorem provers, or human-in-the-loop verification—to support future research as models evolve.

- While the benchmark effectively diagnoses failure modes (e.g., hallucination, guessing, constraint ignoring), the paper does not explore how these insights could be leveraged to improve LLMs’ ability to recognize insolvability—such as through refusal-aware fine-tuning, uncertainty-aware decoding, or integrating external verifiers during inference.

- Typo in line 74, two cannot.

**Questions:**

See weakness

---

### Official Review · Reviewer_JfcQ · 2025-10-28

**Soundness:** 2
**Presentation:** 3
**Contribution:** 1
**Rating:** 2
**Confidence:** 4

**Summary:**

Aiming at the issue that modern large language models (LLMs) perform well on solvable mathematical problems but struggle to recognize mathematical insolvability, this paper proposes MathTrap300—the first benchmark dataset containing 300 unsolvable mathematical problems—and designs a three-stage LLM judge framework (final-answer judgment, midway identification, and problem modification). This framework evaluates the insolvability recognition ability of LLMs from multiple dimensions, and the paper conducts evaluations on 28 state-of-the-art LLMs.

**Strengths:**

1. The paper is clearly expressed, with well-defined motivations and solutions, making it easy to read.
2. The paper proposes a new benchmark, which may be of certain help to subsequent work in this field.

**Weaknesses:**

1. The conclusion of the paper is not novel. It is now common knowledge that large language models (LLMs) struggle to recognize unsolvable problems, so this is hardly a new finding.
2. The paper describes a phenomenon but provides little insight into how to solve the problem, resulting in limited contributions.
3. The experimental section severely lacks detailed descriptions. For example, the authors constructed a dataset of unsolvable problems, but what is the source of the solvable dataset?
4. Attributing the unsolvability of mathematical problems solely to missing conditions and contradictions is too narrow. Currently, the constructed dataset is insufficient in terms of both scale and diversity to meet standard requirements.
5. Table 2 is completely incomprehensible, with no explanations or citations. The authors mentioned in the main text that relevant content would be included in the supplementary materials, but such content is entirely absent from the supplementary materials. It is quite obvious that the paper is still in an unfinished draft stage and is far from being ready for publication; the manuscript generally lacks rigor.

**Questions:**

None.

---

### Official Review · Reviewer_TDWZ · 2025-10-30

**Soundness:** 2
**Presentation:** 3
**Contribution:** 3
**Rating:** 2
**Confidence:** 5

**Summary:**

This paper focuses on evaluating the LLM's mathematical reasoning capability on insolvable math problems, which provides a different view on the LLM math reasoning. Therefore, they propose a Math-Trap300 benchmark for the evaluation that requires LLMs to use deeper math knowledge to conduct reasoning. Among the observations of the LLM outputs, they construct a 3-stage judge framework to provide metrics with more information. Finally, they give the conclusion of the observation that LLMs decreased performance on trap problems and the reasons mainly include hallucination, guessing, and the ignoring of the problem conditions.

**Strengths:**

1.	The core motivation of the paper looks reasonable. This paper focused on evaluating how the performance of LLM on insolvable math problems that be seen as trap problems for the model. With this motivation, they constructed a MathTrap300 benchmark with human-checked rewritten insolvable hard problems.

2.	Compared to previous works using symbol-based or template matching methods, this work tried to provide more evaluation metrics from the failure samples. They leveraged LLM to judge model outputs from 3 different aspects.

3.	With the proposed benchmark and the evaluation method, the authors evaluated 28 SOTA LLMs. Based on the results, the authors further discovered the correlation of the model behaviors and capability of solving insolvable problems.

**Weaknesses:**

1.	Despite requiring human resources to annotate and check the trap problems in the benchmark, the total amount of the current dataset appeared limited with only 300. Meanwhile, the sample number of two types in the benchmark seems extremely imbalance, 274 Contradiction problems and 26 Missing Condition problems. Such a benchmark looks a bit unusual. In the reviewer's opinion, the imbalanced dataset might be  limited to comprehensively show the LLM reasoning ability.

2.	The novelty of this work mainly relies on the design of the benchmark judge process of the model outputs and those given analysis based on the 3 stage judge. The proposed MathTrap300 appears to be an incremental version on the existed MathTrap with harder problems, which may limit the contribution of the paper.

3.	Limited benchmark analysis. It is suggested to provide some deeper analysis and/or draw some charts for the results visualization. For example, in the current MathTrap300, it seems there are various types of questions like “simplify the expression” in figure 2, “If xxx, find m” and “based on several equations to find the result of another equation” in figure 1, they are different types of problems. So, what is the performance of LLM on these different types of problems? More analyses may be required as this paper focused on evaluating the LLM insolvability. To the reivewer, these analyses are important and necessary.

4.	Failure modes analysis mainly relies on human check and another LLM to judge, while it is not ideal that the judger is aligned to human to ensure the reliability and overall assessment. The scalability and comparability of the evaluation results seemed also limited. To position this work as a public benchmark, it is suggested to reduce manual dependence, and  increase automated verifiable signals.

5. Some other issues:

  1). It appears that the URL of dataset and code links to a blank repo, only readme file in there.

  2). Please refine the figures.  The font size is too small and hard to read.

**Questions:**

Please refer to the weaknessespart.

---

### Official Review · Reviewer_B6Ru · 2025-10-30

**Soundness:** 3
**Presentation:** 3
**Contribution:** 2
**Rating:** 4
**Confidence:** 3

**Summary:**

The paper introduces a new dataset, MathTrap300, that contains 300 insolvable, ill-posed math problems. The authors propose an evaluation pipeline that looks at both the final answer and intermediate reasoning to understand models' behavior when prompted with insolvable math problems. Evaluation on 28 LLMs shows a large drop in performance from original problems to their unsolvable counterparts, and studies different failure modes like guessing, hallucination, and constraint-ignoring.

**Strengths:**

1. Quality of the dataset: The paper introduces a high-quality dataset that has been double-verified by PhD-level experts, including paraphrasing to reduce memorization/data contamination.
2. The paper proposes a three-stage LLM judge that captures signals from both the final answer and intermediate reasoning.
3. Comprehensive benchmarking: Evaluation of 28 recent models across chat and reasoning.

**Weaknesses:**

1. The majority of the dataset contains contradiction problems (274 instances) and only 26 are missing condition problems. This can lead to an imbalanced dataset
2. The method of creating unsolvable problems by removing or altering some conditions of the problem has already been explored before[1] [2]. How are they different from the recent literature?




[1]: Fan, Chenrui, et al. "Missing Premise exacerbates Overthinking: Are Reasoning Models losing Critical Thinking Skill?." arXiv preprint arXiv:2504.06514 (2025).
[2]: Xue, Boyang, et al. "Reliablemath: Benchmark of reliable mathematical reasoning on large language models." arXiv preprint arXiv:2507.03133 (2025).

**Questions:**

1. In section 4.1, when the authors mention trap instances, are they referring to MathTrap300 dataset instances?

---

### Note · Authors · 2025-11-12

**Comment:**

We sincerely appreciate the reviewers' time and detailed feedback, which will be valuable in refining our work. After careful consideration, we have decided to withdraw our paper from consideration for the conference. We once again thank the reviewers and the program committee for their thoughtful comments.

**Withdrawal Confirmation:**

I have read and agree with the venue's withdrawal policy on behalf of myself and my co-authors.